# Patterns of Life Lost to Cancers with High Risk of Death in China

**DOI:** 10.3390/ijerph16122175

**Published:** 2019-06-19

**Authors:** Yizhong Yan, Yu Chen, Huaimiao Jia, Jiaming Liu, Yusong Ding, Haixia Wang, Yunhua Hu, Jiaolong Ma, Xianghui Zhang, Shugang Li

**Affiliations:** 1Department of Public Health, Shihezi University School of Medicine, Shihezi 832002, China; erniu19880215@sina.com (Y.Y.); liujiaming@shzu.edu.cn (J.L.); tougaoshzu@126.com (Y.D.); wanghaixiawza@163.com (H.W.); hyh6133@sina.com (Y.H.); jiaojiaolong881202@163.com (J.M.); michaelzhang818@163.com (X.Z.); 2Department of Chronic Diseases, Shihezi Center for Disease Control and Prevention, Shihezi, 832002, China; tougaoshzu1@163.com (Y.C.); tougaoshzu2@163.com (H.J.); 3Department of Pathology and Key Laboratory of Xinjiang Endemic and Ethnic Diseases (Ministry of Education), Shihezi University School of Medicine, Shihezi 832002, China

**Keywords:** patterns, life lost, high risk, death, cancers

## Abstract

To inform public health policy and research, we analyzed the patterns of life lost to cancers and evaluated the cancer burden in China. Based on the published Chinese Cancer Registry Annual Report and related literature in 2013, we calculated the cancer-related mortality and potential years of life lost (PYLL) by age, sex, districts (urban or rural), to describe the patterns of life lost to cancers. The high death-risk cancers in China were lung, liver, stomach, esophageal, colorectal, breast, pancreatic, brain and nervous system, and ovarian cancers, and leukemia. Liver and esophageal cancers were more prominent among males, while breast and colorectal cancers were more prevalent among females. The most obvious differences of mortality between urban and rural areas were found in liver, esophageal, and colorectal cancers. Cancer-related mortality increased significantly after the age of 30 years, and peaking at 70–79 years. The PYLL rate of cancer in urban areas was higher than that in rural areas (21.49 vs. 19.20/1000), and significant regional and gender differences of PYLL ranks can be observed. For people aged over 60 years, cancer PYLL mainly came from lung, stomach, and esophageal cancers; for middle-aged people, it was mainly induced by liver, colorectal, and female reproductive systems’ cancers; and for those under 30 years, life lost to cancer was mainly caused by leukemia and brain, nervous system cancers. Moreover, disparities in age distribution of PYLL from different regions and sexes can be found. In short, three categories of people, including those in urban areas, males and people over 60 years, were suffering from more serious cancer deaths and life lost. These variations pose considerable challenges for the Chinese health care system, and comprehensive measures are required for cancer prevention and treatment.

## 1. Introduction

According to the data released by the International Agency for Research on Cancer (IARC), about 9.6 million deaths occurred from cancer worldwide, in 2018 [1], which was 17.1%, 26.3%, and 43.3% higher than those reported in 2012, 2008, and 2002, respectively (8.2, 7.6, and 6.4 million, respectively) [2,3,4]. Although some measures such as early detection and cancer prevention, screening, and treatment have been implemented, cancer deaths have not been controlled adequately [1,2,3,4], and have resulted in a serious life lost, especially in low- and middle-income countries (LMIC) [1,2,3,4,5]. It was estimated that cancer deaths will increase in Asia from 4.1 million in 2008 to 7.5 million in 2030 [4]. Furthermore, cancer is also expected to cause disability and shorten life expectancy, and accordingly increase social burdens substantially [6]. To control cancers holistically, we need to not only understand what happens before the onset of cancers, but also the state after the disease is diagnosed, especially because cancers are not yet well treated at present. Therefore, studies on the evaluation of cancer burden, such as the life lost, are important for developing appropriate measures for cancer prevention and prognosis. Such studies would provide empirical evidence for governments to allocate health resources legitimately.

The population of China surpassed nearly 1.3 billion, accounting for about one fifth of the world’s total population, a large population base whose disease epidemics, distribution and burden will have a significant impact on the world. Furthermore, obvious differences existed in regional characteristics, for example between urban and rural areas, especially in terms of factors such as culture, economy, and medical services [7]. Because of the size of the agricultural population [8], a weak health care system [9], growing environmental pollution [10], and aging of the population [11], China has a greater cancer- related mortality as compared with developed countries [12,13]. However, despite the increasing cancer burden in China [14], we were unable to identify more reports on this topic. Potential years of life lost (PYLL) is an important measure that emphasizes the impact of “premature death” by eliminating the influence of death age composition on life expectancy loss; it is more sensitive than mortality in measuring the mortality level in the same period or in different years of the same disease [15]. According to the PYLL, we can know which of several kinds of cancers causes the most serious “premature death”, so as to determine the focus of prevention and treatment.

In the present study, using the latest updated and published data from the National Cancer Center (NCC) of China, we aimed to describe the pattern of life lost to cancer and evaluate the cancer burden of China by calculating a crude mortality rate (CMR), standardized mortality rate (SMR), potential years of life lost (PYLL), PYLL rate (PYLLR), and standardized PYLL rate (SPYLLR). Furthermore, to define the age of high cancer burden, age distribution charts were developed. Therefore, the combination of total death and premature death will provide a comprehensive description of the life lost caused by cancer. These findings could provide evidence for developing policies for cancer prevention and control in China, as well as act as a reference for future cancer research.

## 2. Materials and Methods

### 2.1. Data Sources

Cancer-related mortality by gender, age, and districts (urban or rural), and population data for 2013 were derived from the published “Chinese Cancer Registry Annual Report, 2016” and the paper “Report of Cancer Incidence and Mortality in China, 2013” [16,17]. The annual report of 2016 was the latest annual report which included the largest number of cancer registries, and the quality of registration data was high, which was representative of the cancer epidemic in China [16]. Additionally, we analyzed the ranking of different cancers by mortality, to identify high-death-risk cancers in China by age, sex, and region.

The National Cancer Center, based on the requirements of the Chinese Cancer Registration Manual [18] and the International Agency for Research on Cancer (IARC) [19,20,21], reviewed the data reported by all the registration points in China. And the reliability, completeness, validity and timeliness of the data were assessed by some indicators, such as the proportion of pathological diagnosis (MV%), the proportion of only death certificate (DCO%) and mortality/incidence ratio (M/I). Finally, 255 registries were incorporated into the registration office in 2013, covering 226,494,490 people (114,860,339 for males, 111,634,151 for females; 111,595,772 for urban areas, 114,898,718 for rural areas), and accounting for 16.65% of the population at the end of the year (16.72% for males and 17.09% for females; 16.77% for urban areas and 17.04% for rural areas) [16]. In 2013, the MV% was 67. 89%, DCO% was 1.81%, and M/I was 0.62 [13], all of them met the target. 

### 2.2. Analysis Index

According to the international classification of diseases (ICD-10) [22], the CMR, SMR, constituent ratio (%), PYLL, PYLLR and SPYLLR were calculated for all cancers (C00-C96):

PYLL

This is an analysis index considering the age of death and the detriment of disease to human beings. It can measure the extent of harm from a death cause to a certain age group, and is often used to evaluate the risk of a disease causing “premature death” [23,24]. 

Premature death, also known as advanced death, refers to death occurring before the average life span, which is evaluated by the difference between the actual age and the life expectancy, that is, the number of years of life lost in a certain cause resulting in the loss of life expectancy. The PYLL formula is as follows: [25]
(1)$$PYLL=\sum \left(L-a_{i}\right)\times d_{i}$$

PYLLR

PYLL rate, PYLL per 1000 person, it is the loss of life per 1000 person in the target population, and is available for comparison:(2)PYLLR=PYLL/N×1000‰

SPYLLR

If the population composition is different, the standardized PYLL rate should be calculated before comparison: (3)SPYLLR=∑(PYLL×Correction coefficient)/N×1000‰

In the formula (1)–(3):

L is the life expectancy based on the Sixth National Population Census of China in 2010 [26,27], which calculated life expectancy in terms of urban and rural areas, males and females, respectively. All the areas: 78.16 years for all the population, 75.79 for males, 80.77 for females; urban areas: 82.22 years for all the population, 80.13 for males, 84.43 for females; rural areas: 75.87 years for all the population, 73.30 for males, 78.76 for females.

a_i_ is the median of age group *i*: 0.5 for 0-group, 2.5 for 1–4 group, 7 for 5–9 group, 12 for 10–14 group, 17 for 15–19 group, 22 for 20–24 group, 27 for 25–29 group, 32 for 30–34 group, 37 for 35–39 group, 42 for 40–44 group, 47 for 45–49 group, 52 for 50–54 group, 57 for 55–59 group, 62 for 60–64 group, 67 for 65–69 group, 72 for 70–74 group, 77 for 75–79 group, 82 for 80–84 group, 87 for 87–89 group. This age grouping method was based on the Chinese Cancer Registry Annual Report [16], and it calculated the related results in this way.

d_i_ represents the number of deaths in age group *i*. Age grouping and the data of each group also came from the Chinese Cancer Registry Annual Report [16].

Correction coefficient = (Pir/Nr) ÷ (Pi/N): Pir/Nr is the age composition of the standard population from the Sixth National Population Census of China in 2010 [26]; Pi/N is the age composition of the observation population from the Chinese Cancer Registry Annual Report [16].

N is the total population of premature deaths. PYLL judges premature death according to the difference between L and a_i_. If the difference is greater than 0, it is believed that premature death has occurred. Therefore, according to L of different regions and sexes, the age threshold of premature death based on life expectancy in this study is as follows: all the areas: 77 years (median of 75–79 group)for all the population, 72 (median of 70–74 group) for males, 77 years (median of 75–79 group) for females; urban areas: 82 years (median of 80–84 group) for all the population, 77 years (median of 75–79 group) for males, 82 years (median of 80–84 group) for females; rural areas: 72 (median of 70–74 group) years for all the population, 72 (median of 70–74 group) for males, 77 years (median of 75–79 group) for females.

Finally, in order to describe the age group more concisely, we merged them according to the World Health Organization (WHO) age grouping method used to estimate cancer burden (0–4, 5–14, 15–29, 30–44, 45–59, 60–69, 70–79, 80+) [28], and PYLL is additive [23].

### 2.3. High Death-Risk Cancers

We ranked nationwide cancer-related mortality rates by district (urban or rural) and sex, and then selected the top 10 as the high death-risk cancers in China. Subsequently, we conducted a detailed analysis of the life loss patterns caused by these high death-risk cancers.

### 2.4. Statistical Analysis

Using Excel 2016 (Microsoft Corporation, Redmond, Washington, USA) and SPSS 25.0 software (International Business Machines Corporation, Among, New York, USA) to collate and analyze the data. The standardized mortality rate of China (SMRC) was calculated according to the standard population age composition of the national census in 2010 [26], and the standardized mortality rate of the world (SMRW) was calulated according to the age constitution of Segi’s world standard population [29]. Chi-square test was used to compare the rates between urban and rural, males and females. The test level was *P* = 0.05.

## 3. Results

### 3.1. Mortality and Ranks of Cancers

In 2013, 2.229 million individuals died of cancer in China, with a CMR of 163.8/10^5^, including 1.406 million males and 823,000 females (CMRs of 201.7/10^5^ and 124.1/10^5^, respectively). The CMR for lung cancer was the highest, at 46.9/10^5^, followed by those for liver, stomach, esophageal, colorectal, breast, pancreatic, brain and nervous system, and ovarian cancers, and leukemia. Similarly, for males and females, the highest CMR was for lung cancer, at 62.9/10^5^ and 30.5/10^5^, respectively. It was followed by those for liver cancer, stomach cancer, esophageal cancer, colorectal cancer, pancreatic cancer, brain and nervous system cancer, leukemia, lymphoma, and prostate cancer in males. In females, the CMR for lung cancer was followed by those for stomach, liver, colorectal, breast, esophageal, pancreatic, cervical, brain and nervous system cancers, and leukemia (Table 1). 

In urban areas, 1.181 million individuals died of cancer, with a CMR of 161.5/10^5^, including 736,000 males and 445,000 females (CMRs of 197.2/10^5^ and 124.3/10^5^, respectively). The CMR for lung cancer was the highest, at 50.5/10^5^, followed by those for liver cancer, stomach cancer, colorectal cancer, breast cancer, esophageal cancer, pancreatic cancer, prostate cancer, leukemia, and cervical cancer. Similarly, for both sexes, the highest CMR was for lung cancer (67.8/10^5^ and 32.9/10^5^, respectively). In males, it was followed by those for liver cancer, stomach cancer, esophageal cancer, colorectal cancer, pancreatic cancer, brain and nervous system cancer, leukemia, lymphoma, and prostate cancer. In females, the CMR for lung cancer was followed by those for stomach, liver, colorectal, breast, esophageal, pancreatic, cervical, and brain and nervous system cancers, and leukemia (Table 1).

In rural areas, 1.048 million individuals died of cancer, with a CMR of 166.6/10^5^, including 670,000 males and 378,000 females (CMRs of 206.9/10^5^ and 123.8/10^5^, respectively). The CMR for lung cancer was the highest, at 43.5/10^5^, followed by those for stomach, liver, esophageal, colorectal, breast, pancreatic, brain and nervous system, and ovarian cancers, and leukemia. Additionally, for both males and females, the highest CMR was for lung cancer, at 58.1/10^5^ and 28.1/10^5^, respectively. It was followed by those for liver cancer, stomach cancer, esophageal cancer, colorectal cancer, pancreatic cancer, brain and nervous system cancer, leukemia, lymphoma, and prostate cancer in males. In females, the CMR for lung cancer was followed by those for stomach, liver, esophageal, colorectal, breast, pancreatic, cervical, and brain and nervous system cancers, and leukemia (Table 1).

### 3.2. Age Distribution of High Death-Risk Cancers

A significant increase in cancer deaths was observed after the age of 30 years, and peaking at 70–79 years. At the age of 0–29 years, the majority of cancer-related deaths were caused by leukemia, brain and nervous system cancer, and liver cancer. At the age of 30–44 years, the most deaths were caused by liver cancer, which increased gradually, and peaking at 45–59 years. Simultaneously, the number of deaths caused respectively by breast, brain and nervous system, and cervical cancers, and leukemia also reached their peaks in this age group. In other age groups, the most deaths were caused by lung cancer, and peaking at 70–79 years. Additionally, the maximum number of deaths caused respectively by stomach, esophageal, colorectal, and pancreatic cancers also appeared at this age (Figure 1c). The peak values for deaths caused by the top 10 cancers by ranking mortality in urban areas appeared in the following order of age groups: 45–59 years (liver, breast, and ovarian cancer), 70–79 years (lung, stomach, esophageal, and pancreatic cancers, and leukemia), and 80+ years (colorectal and prostate cancer) (Figure 1a). The corresponding distribution in rural areas was as follows: 45–59 years (liver, breast, brain and nervous system, and cervical cancers, and leukemia), 70–79 (lung, stomach, esophageal, colorectal, and pancreatic cancers) (Figure 1b).

### 3.3. Potential Years of Life Lost (PYLL) and Ranks of High Death-Risk Cancers

In 2013, the cancer-related PYLL in China was 4.85 million person-years, PYLLR was 21.93/1000, and SPYLLR was 19.46/1000, with 2.64 million person-years, 23.98/1000, and 21.78/1000 for males, respectively; and 2.06 million person-years, 18.98/1000, and 16.87/1,000 for females, respectively. Figure 2c shows the top 10 PYLL rates were observed for lung, liver, stomach, breast, colorectal, esophagus, leukemia, brain and nervous system, cervix and pancreas cancers. Among males, they were observed for lung, liver, stomach, esophageal, and colorectal cancers etc, while among females, they were observed for lung, breast, liver, stomach, and colorectal cancers etc. Of all the subjects, lung cancer had the highest PYLL rate, 4.95/1000 for both, 5.75/1000 for males and 3.71/1000 for females (Table 2).

In urban areas, the cancer-related PYLL was 2.94 million person-years, PYLLR was 26.63/1000, and SPYLLR was 21.49/1000, with 1.64 million person-years, 29.98/1000, and 23.86/1000 for males, respectively; and 1.23 million person-years, 22.57/1000, and 18.36/1,000 for females, respectively. Figure 2a shows the top 10 PYLL rates were observed for lung, liver, stomach, breast, colorectal, esophageal, leukemia, pancreatic, ovarian and prostate cancers. In males, they were observed for lung, liver, stomach, esophageal and colorectal cancers etc.; in females, they were observed for lung, breast, stomach, liver, and colorectal cancers etc. In the urban population, the PYLL rates of lung cancer were the highest, 6.47/1000 for both, 7.96/ 1000 for males and 4.56/1000 for females (Table 2).

In rural areas, the PYLL for cancers was 2.21 million person-years, PYLLR was 20.02/1000, and SPYLLR was 19.20/1000, with 1.18 million person-years, 20.87/1000, and 20.05/1000 for males, respectively; and 0.95 million person-years, 17.26/1000, and 17.09/1,000 for females, respectively. Figure 2b shows the top 10 PYLL rates were observed for lung, liver, stomach, esophageal, breast, colorectal, leukemia, brain and nervous system, cervical and pancreatic cancers. In males, they were observed for liver, lung, stomach, esophageal and leukemia cancers etc; in females, they were observed for lung, liver, breast, stomach, and esophagus cancers etc. The PYLL rate caused by lung cancer was the highest in rural population, was 4.27/1000, which was also the highest in females, was 3.33/1000, but the highest PYLL rate in males was liver cancer, was 5.45/1000 (Table 2).

### 3.4. PYLL Age Distribution of High Death-Risk Cancers

Figure 3 shows, among Chinese residents, premature death caused by cancers was the most serious among 60–69 years, and lung cancer was the leading cause for both males and females, and the highest status among males was more prominent. The PYLLs were higher for lung, stomach, esophageal, and pancreatic cancers in individuals aged 60–69 and 70–79 years; and those for liver, colorectal, breast, and cervical cancers were higher for individuals aged 30–44 and 45–59 years. For those younger than 30 years, the main cancers causing premature death were leukemia and brain, nervous system cancers. It is worth mentioning that the liver PYLL was also in a higher position at the age of 15–29 years, and when the population entered 70–79 years, the liver PYLL decreased the most comparison with lung and stomach cancers (from 9.57 to 3.79/1000) (Figure 3c). Among males over 60 years, categories of cancer with the highest PYLL were the same as that of the total population, except for colorectal cancer, whose highest PYLL was also at 60–69 years in males. The highest PYLL of liver cancer in males was at 45–59 years. In addition, leukemia was the primary cause of premature death in males aged 0–29 years, especially in 0–4 years (Figure 3a). Types of cancer with the highest PYLL in females over 60 years were similar to those in males, but unlike, the liver cancer PYLL of females was also the highest at the group older than 60. The highest PYLLs for breast and cervical cancers were observed at 45–59 years, but that for leukemia was the highest at 0–4 years (Figure 3b).

Figure 4 shows that the premature death from cancers among urban residents was most severe between 70 and 79 years. The PYLLs of lung, stomach, esophageal, colorectal and pancreatic cancers were higher in people over 60 years, and those of liver, breast and ovarian cancers were higher in middle-aged people. Leukemia was the leading cause of premature death in people under 30 years, meanwhile liver cancer also played an important role, especially in those aged 15–29 (Figure 4e). Among males over 60 years, the highest PYLL cancers were the same as those of the total population. The highest PYLL of liver cancer in males was from 45–59 years. Leukemia and brain, nervous system cancers were the primary cause of premature death in males aged 0–29 years, especially in 0–4 years (Figure 4a). One significant difference between males and females was that the highest PYLL of liver cancer in females was observed at 70–79 years. The PYLLs from breast, cervical and ovarian cancers were the highest aged 45–59 years. The main cancers that induce premature death aged 0–29 was the brain, nervous system. In addition, it should be mentioned that esophageal cancer induced much lower PYLL in females of all ages (Figure 4c).

Figure 4 shows that, in rural areas, the premature death from cancers was the most serious at 60–69 years, earlier than urban areas. The PYLLs for lung, stomach, esophageal, and pancreatic cancers were higher among individuals over 60 years; and those for liver, breast, and cervical cancers were higher among individuals aged 45–59 years. The PYLL for leukemia was the highest among those aged 0–4 years (Figure 4f). In males, the PYLLs for lung, stomach, esophageal, colorectal, pancreatic, and prostate cancers were higher among individuals aged 60–74 years; and that for liver cancer peaked at 45–59 years. For the person aged <30 years, the main cancers induced premature death were leukemia and brain, nervous system cancers. (Figure 4b). In females, the PYLLs for lung, stomach, esophageal, and pancreatic cancers were higher among individuals aged 60–79 years, but unlike males, the peak age of liver and colorectal cancers PYLL has been delayed, mainly also occurring in people over 60 years. The highest PYLLs for breast and cervical cancers were observed at 45–59 years, and those for leukemia at 0–4 years (Figure 4d).

## 4. Discussion

In the present study, PYLL was used to analyze the phenomenon of premature death caused by cancer, and the life lost from cancer among Chinese residents has been fully presented. We conclude that, among the cancers frequently occurring, lung, liver, stomach, breast and colorectal cancers were the most serious causes of life lost in China, and significant regional, gender and age differences existed in our results. The life lost induced by cancer in urban areas and males were even worse than in rural areas and females, and all of them mainly occurred among people over 60 years old. It was thus clear that significant diversities existed in the patterns of life lost of cancer among Chinese residents. Research on and clarification of these differences may be helpful and instructive for cancer prevention and control.

The rankings by cancer deaths were similar to the findings pertaining to PYLLs, but a significant difference was observed among females in urban areas, who exhibited a higher CMR (second rank) but lower PYLL (fifth rank) for colorectal cancer. However, the circumstances of breast cancer was the opposite (fifth rank for CMR, second rank for PYLL). This phenomenon can be explained by another finding of the present study, that deaths caused by colorectal cancer were common among individuals aged 80+ years, while those caused by breast cancer mainly occurred among individuals aged 45–59 years. As the latter were younger, they exhibited a higher PYLL. It can be seen that female breast cancer was a kind of “younger cancer”, and the peak age of death of Chinese female breast cancer was younger than that of other countries. For example, American female breast cancer patients died mostly between 55 and 64 years of age [30], while Korean women who were also in Asia mostly died at 60–64 years [31], with the peak behind. So the premature death caused by breast cancer in China may be more grievous, which should be paid more attention to.

Additionally, the overall premature death of cancer in urban areas was more serious than that in rural areas, especially lung cancer. However, the conclusions of PYLL about urban and rural reported by Zhang et al. [32] was inconsistent with ours. The reason was that the algorithm of PYLL was different, and they used the same life expectancy in urban and rural when calculating PYLL; however, life expectancy in urban areas was higher than that in rural areas. Based on this, we have made improvements and adopted different life expectancy for different groups of people, which was more reasonable and could accurately reflect the premature death caused by cancer. Besides, the PYLLs of liver, stomach and esophageal cancers in rural were significantly higher than in urban areas, as observed in other studies [33]. Furthermore, the life lost of cancer among individuals aged 0–14 years, 15–59 years, and 60 years was attributed to leukemia, liver cancer, and lung cancer, respectively. This finding indirectly suggests that leukemia, and liver and lung cancers, were major cancers that had a greater impact on the life lost among Chinese residents [34,35].

Worldwide, lung cancer was reported as the leading cause of cancer deaths among males and the second leading cause among females [2]. In the present study, lung-cancer-related mortality was the highest in both sexes. For males, this may be related to a higher smoking rate [36]; while, for females, although the smoking rate was low, the passive smoking rate of non-smoking females in China was as high as 71.6%, which is substantially higher than that in other countries [37]. Notably, the lung cancer PYLL in males from rural areas was the second highest after liver cancer, but among other populations, it was the highest. Additionally, the mortality and life lost to liver cancer in rural areas were more serious than those in urban areas. Other studies also reported that the burden of liver cancer was higher in males than in females, especially in developing rather than developed countries [2]. Furthermore, the highest mortality related to liver cancer was reported in the western region of China, which may be related to the low vaccination rate for hepatitis B [38]. According to some studies, the prevalence rate of hepatitis B in the eastern and central regions of China had decreased significantly, while that in the western region remained high [39]. The hepatitis B vaccine (HBV) is 95% effective in preventing hepatitis B infections, and sustaining a high coverage through expanded immunization programs is vital for ensuring a steady decline in the incidence of liver cancer in future [40,41]. All of the above reasons suggest that the death and life lost to liver cancer were more serious in less developed areas, the HBV coverage rate should be increased in these areas, to reduce the burden of liver cancer effectively.

With the development of the social economy, large changes in lifestyle, and rapid aging of the population, mortality related to colorectal cancer in China was reported to be on the rise, especially in urban areas [38]. In Global Cancer (Globocan) 2018, colorectal cancer was reported as the fifth most common cancer that caused death, being the fourth most common among males and the third most common among females [1]. In the present study, colorectal cancer was the second leading cause of death among females in urban areas, after lung cancer. Evidently, urban females may be more likely to die from colorectal cancer. Another study showed that treatment at early clinical stages was associated with excellent prognosis, as follows: the 5-year survival for Stage 1 and 2 colorectal cancer exceed 90% and 80%, respectively; it ranged from 30% to 60% for Stage 3; and was less than 10% for Stage 4 in Japan [42,43]. In the United States and other areas with a high incidence of colorectal cancer, the mortality has declined since 1985, mainly due to the spread of screening and improvement of treatment. Accordingly, China needs to improve public health care infrastructure to support Fecal Occult Blood Test (FOBT) screening, colonoscopy, and staging and treatment of colorectal cancer, especially targeting urban females.

The death and burden related to esophageal cancer in China also showed significant regional and gender differences. The PYLL for esophageal cancer was higher among males, while it did not appear in the top five ranks among females. This finding may be explained by the fact that males are more likely to have several unhealthy habits such as smoking and alcohol abuse. In the Chinese population, about half of esophageal-cancer-related deaths were attributable to smoking, alcohol drinking, and low vegetable and fruit intake, with smoking and alcohol drinking being responsible for about 30% of the mortality related to esophageal cancer [44]. Therefore, China should redouble its efforts to reduce tobacco smoking to ensure a decline in the death and disease burden related to esophageal cancer, especially in rural areas and among males. 

Although the present study only described the current situation, the above description of existing and emerging patterns related to cancer in China calls for the development of varied interventions to reduce the disease burden. Such efforts should include balanced investment in awareness and educational efforts targeting the general population and primary care practitioners, particularly focusing on the primary prevention of hepatitis B and human papilloma virus infections, tobacco control measures, and promotion of physical activity and healthy diet. 

## 5. Conclusions

In sum, it was meaningful to understand the patterns of cancer-related death and life lost, in that this information could be used to develop cancer prevention and treatment measures for different age groups, genders, and regions. We conclude that lung, liver, stomach, breast and colorectal cancers were the main cancers causing premature death in China. These cancers were also at the forefront of the PYLL ranking in urban and rural areas, but for rural residents, besides these cancers, the impact of esophageal cancer could not be ignored. For males and females, in addition to paying close attention to lung cancer, separate attention should be given to liver and breast cancers. People aged over 60 years were a concentrated age group with a large number of cancers causing life lost. Based on these findings, the relevant departments should assign importance to high-risk populations and cancers, to effectively reduce the burden of cancer and improve quality of life.

## Figures and Tables

**Figure 1 ijerph-16-02175-f001:**
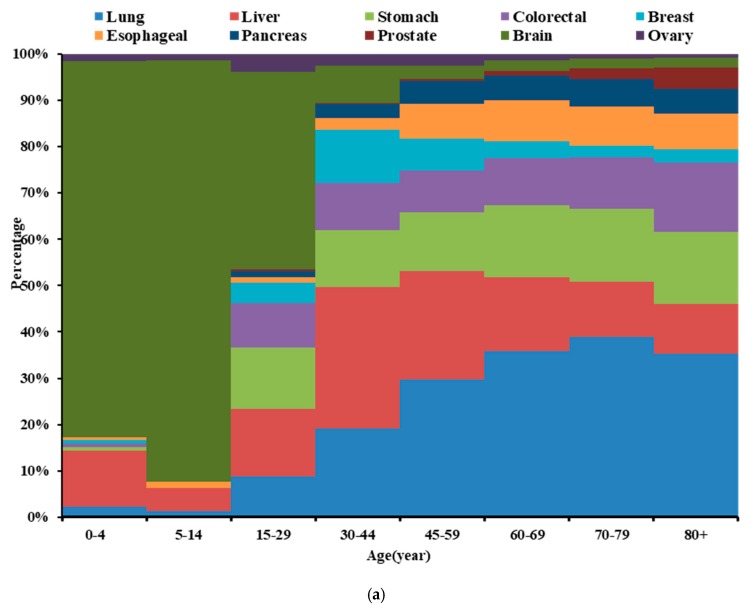
Deaths age distribution of high-death-risk cancers, China, 2013. (**a**) urban; (**b**) rural; (**c**) both.

**Figure 2 ijerph-16-02175-f002:**
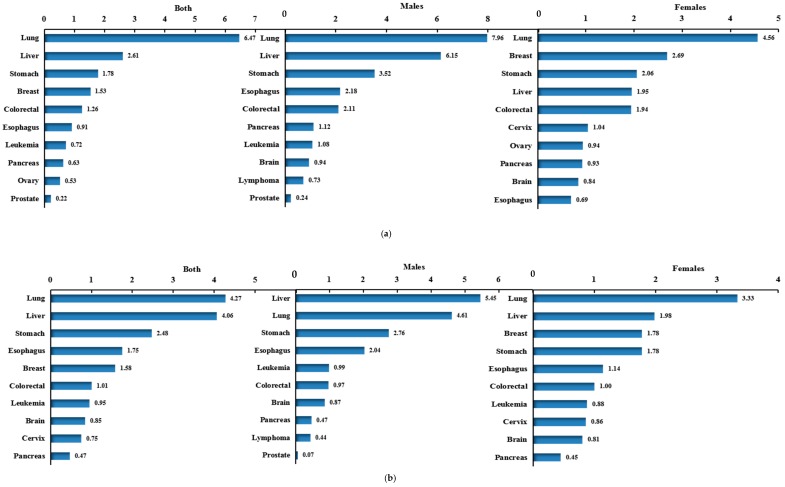
Ranks of high-death-risk cancers PYLL rate in China by region and gender, 2013. (**a**) urban; (**b**) rural; (**c**) both.

**Figure 3 ijerph-16-02175-f003:**
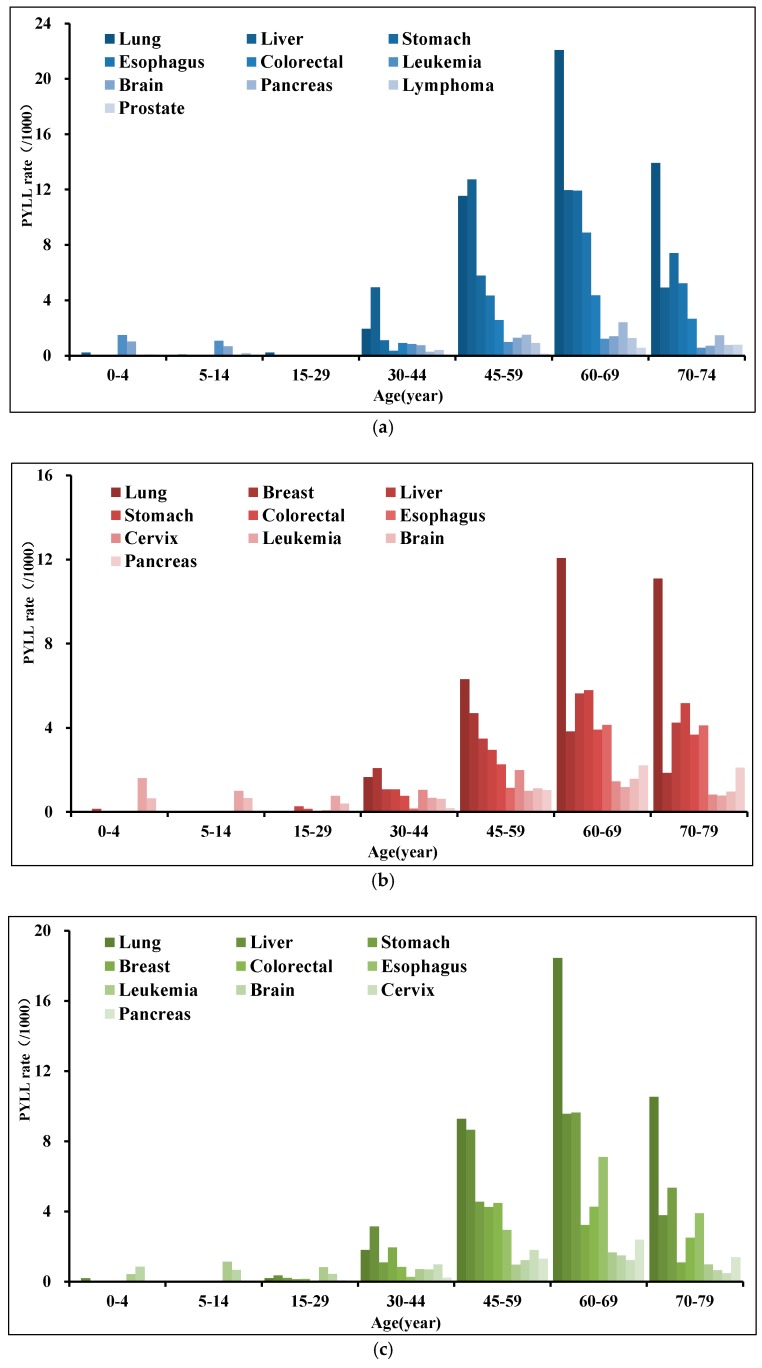
PYLL age distribution of high-death-risk cancers in China, 2013. (**a**) males; (**b**) females; (**c**) both.

**Figure 4 ijerph-16-02175-f004:**
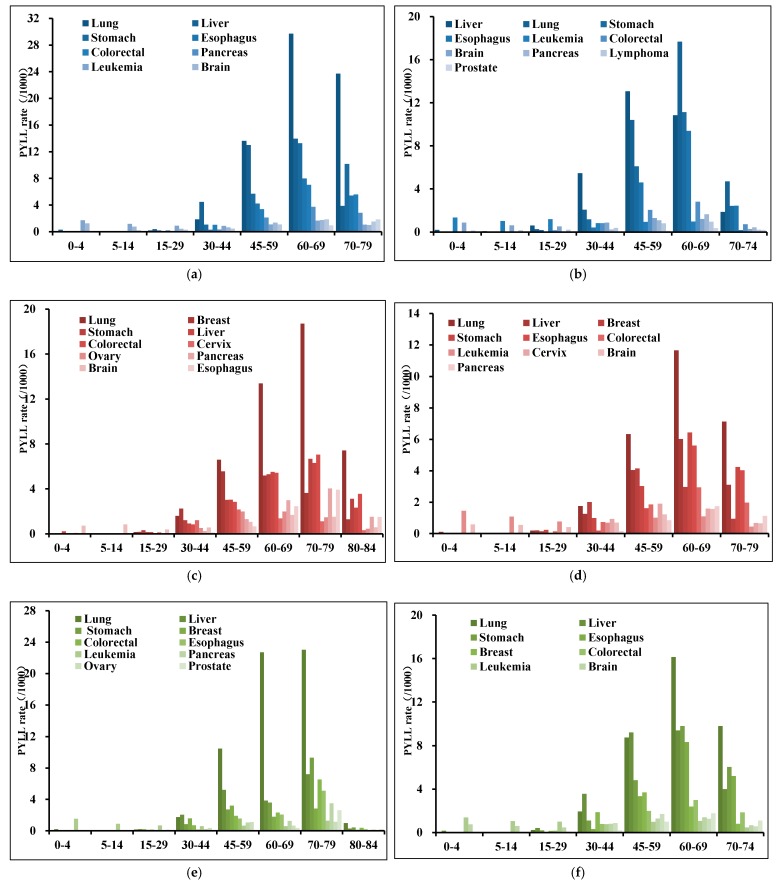
PYLL age distribution of high-death-risk cancers in urban and rural areas, 2013. (**a**) urban males; (**b**) rural males; (**c**) urban females; (**d**) rural females; (**e**) urban both; (**f**) rural both.

**Table 1 ijerph-16-02175-t001:** Mortality and ranks of cancers in China, 2013(/10^5^).

Site	Sex	All Areas					Urban Areas					Rural Areas				
		CMR *	SM RC	SM RW	%	R	CMR*	SM RC	SM RW	%	R	CMR *	SM RC	SM RW	%	R
Esophagus	M	23.0	14.9	15.0	10.5	4	16.8	10.1	10.2	7.5	5	29.0	20.2	20.2	13.5	4
	F	10.1	5.6	5.6	7.6	6	6.2	3.1	3.1	4.5	7	13.9	8.4	8.3	11.0	4
	B	16.6	10.2	10.2	9.4	4	11.5	6.5	6.6	6.4	6	21.6	14.2	14.2	12.6	4
stomach	M	32.9	21.4	21.3	15.0	3	29.5	17.8	17.7	13.2	3	36.0	25.4	25.2	16.8	3
	F	15.1	8.8	8.6	11.4	2	13.6	7.4	7.2	9.8	3	16.6	10.3	10.2	13.2	2
	B	24.1	14.9	14.8	13.7	3	21.6	12.4	12.2	11.9	3	26.6	17.7	17.5	15.5	2
Colorectal	M	14.6	9.4	11.1	6.7	5	18.5	10.9	12.9	8.3	4	10.9	7.7	8.9	5.0	5
	F	11.4	6.4	7.5	8.7	4	14.4	7.4	8.8	10.4	2	8.4	5.3	6.0	6.7	5
	B	15.2	9.0	10.6	7.3	5	16.5	9.1	10.7	9.1	4	9.7	6.5	7.4	5.6	5
Liver	M	35.8	24.4	24.0	16.4	2	33.3	21.2	20.9	14.9	2	38.3	27.8	27.3	17.8	2
	F	13.3	7.9	7.8	10.0	3	12.3	6.7	6.7	8.8	4	14.3	9.2	9.1	11.3	3
	B	24.7	16.1	15.9	14.0	2	22.8	13.9	13.7	12.6	2	26.5	18.5	18.2	15.5	3
Pancreas	M	6.9	4.6	4.5	3.2	6	8.7	5.2	5.2	3.9	6	5.4	3.8	3.8	2.5	6
	F	5.6	3.2	3.1	4.2	7	7.0	3.7	3.6	5.0	6	4.2	2.6	2.6	3.3	7
	B	6.3	3.9	3.8	3.6	7	7.8	4.4	4.4	4.3	7	4.8	3.2	3.2	2.8	7
Lung	M	62.9	40.7	40.6	28.7	1	67.8	40.5	40.4	30.3	1	58.1	40.8	40.6	27.1	1
	F	30.5	17.6	17.3	23.1	1	32.9	17.4	17.1	23.7	1	28.1	17.8	17.6	22.4	1
	B	46.9	28.8	28.6	26.6	1	50.5	28.5	28.3	27.8	1	43.5	29.0	28.8	25.4	1
Breast	M	-	-	-	-	-	-	-	-	-	-	-	-	-	-	-
	F	10.1	6.6	6.4	7.6	5	11.9	7.2	7.0	8.6	5	8.2	5.8	5.6	6.5	6
	B	10.1	6.6	6.4	2.9	6	11.9	7.2	7.0	3.3	5	8.2	5.8	5.6	2.4	6
Ovary	M	-	-	-	-	-	-	-	-	-	-	-	-	-	-	-
	F	4.1	2.8	2.7	3.1	8	4.0	2.6	2.4	2.9	9	4.2	3.0	2.9	3.4	
	B	4.1	2.8	2.7	1.2	9	4.0	2.6	2.4	1.1	13	4.2	3.0	2.9	1.2	9
Cervix	M	-	-		-	-	-	-	-	-	-	-	-	-	-	-
	F	3.3	2.2	2.1	2.5	11	4.2	2.5	2.5	3.0	8	2.5	1.7	1.7	2.0	11
	B	3.3	2.2	2.1	0.9	13	4.2	2.5	2.5	1.2	10	2.5	1.7	1.7	0.7	12
Prostate	M	4.0	2.4	2.4	1.8	10	5.7	2.9	3.0	2.5	7	2.4	1.6	1.6	1.1	10
	F	-	-	-	-	-	-	-	-	-	-	-	-	-	-	-
	B	4.0	2.3	2.4	1.2	11	5.7	2.9	3.0	1.6	8	2.4	1.6	1.6	0.7	13
Brain, nervous system	M	4.7	3.5	3.5	2.1	7	4.4	3.1	3.1	2.0	10	5.0	3.9	3.8	2.3	7
	F	3.9	2.7	2.6	2.9	9	3.8	2.5	2.5	2.7	10	3.9	2.9	2.8	3.1	9
	B	4.3	3.1	3.0	2.4	8	4.1	2.8	2.8	2.3	11	4.5	3.4	3.3	2.6	8
Lymphoma	M	4.0	2.8	3.1	1.8	9	4.8	3.1	3.4	2.2	9	3.3	2.5	2.6	1.6	9
	F	2.8	1.7	1.9	2.1	12	3.3	1.9	2.1	2.4	12	2.2	1.5	1.6	1.7	12
	B	3.4	2.2	2.5	2.0	12	4.1	2.5	2.8	2.2	12	2.7	1.9	2.1	1.6	11
Leukemia	M	4.6	3.6	3.8	2.0	8	4.9	3.6	3.9	2.2	8	4.3	3.6	3.8	2.1	8
	F	3.5	2.6	2.8	2.6	10	3.7	2.6	2.8	2.6	11	3.3	2.6	2.7	2.6	10
	B	4.0	3.1	3.3	2.3	10	4.3	3.1	3.3	2.4	9	3.8	3.1	3.3	2.2	10

Notes: M: males; F: females; B: both of the males and females. CMR: crude mortality rate. SMRC: standardized mortality rate of China. SMRW: standardized mortality rate of the world. CMR, SMRC and SMRW are “per 100,000 persons”. %: constituent ratio. R: ranks. * these data were from references [16].

**Table 2 ijerph-16-02175-t002:** Potential years of life lost (PYLL) and ranks of the high death-risk cancers in China, 2013(/10^3^).

R^1^	Sex	All Areas				Urban Areas				Rural Areas			
		Site	PR	SPR	R^2^	Site	PR	SPR	R^2^	Site	PR	SPR	R^2^
1	M	Lung	5.75	5.17	1	Lung	7.96	5.99	1	Lung	4.61	4.41	2
	F	Lung	3.71	3.20	1	Lung	4.56	3.50	1	Lung	3.33	3.35	1
	B	Lung	4.95	4.26	1	Lung	6.47	4.94	1	Lung	4.27	4.07	1
2	M	Liver	5.66	5.07	2	Liver	6.15	5.11	2	Liver	5.45	5.20	1
	F	Stomach	1.86	1.63	4	Colorectal	1.94	1.52	5	Stomach	1.78	1.72	4
	B	Liver	4.00	3.57	2	Liver	2.61	2.20	2	Stomach	2.48	2.37	3
3	M	Stomach	3.02	2.73	3	Stomach	3.52	2.68	3	Stomach	2.76	2.64	3
	F	Liver	1.91	1.68	3	Stomach	2.06	1.67	3	Liver	1.98	1.94	2
	B	Stomach	2.57	2.23	3	Stomach	1.78	1.42	3	Liver	4.06	3.87	2
4	M	Esophageal	2.11	1.88	4	Colorectal	2.11	1.67	5	Esophagus	2.04	1.95	4
	F	Colorectal	1.34	1.17	5	Liver	1.95	1.55	4	Esophageal	1.14	1.22	5
	B	Esophageal	1.63	1.38	6	Colorectal	1.26	1.01	5	Esophageal	1.75	1.67	4
5	M	Colorectal	1.37	1.24	5	Esophageal	2.18	1.63	4	Colorectal	0.97	0.93	6
	F	Breast	2.14	1.93	2	Breast	2.69	2.28	2	Colorectal	1.00	1.00	6
	B	Colorectal	1.83	1.60	5	Breast	1.53	1.33	4	Colorectal	1.01	0.96	6
6	M	Pancreas	0.70	0.63	8	Pancreas	1.12	0.85	6	Pancreas	0.47	0.44	8
	F	Esophagus	0.94	0.79	6	Pancreas	0.93	0.70	8	Breast	1.78	1.66	3
	B	Breast	1.88	1.70	4	Esophageal	0.91	0.69	6	Breast	1.58	1.50	5
7	M	Brain	0.89	0.85	7	Prostate	0.24	0.17	10	Brain	0.87	0.86	7
	F	Pancreas	0.62	0.53	10	Esophageal	0.69	0.51	10	Pancreas	0.45	0.46	10
	B	Pancreas	0.67	0.57	10	Pancreas	0.63	0.49	8	Pancreas	0.47	0.45	10
8	M	Leukemia	1.01	0.99	6	Leukemia	1.08	1.04	7	Leukemia	0.99	0.99	5
	F	Cervix	0.93	0.85	7	Ovary	0.94	0.78	7	Cervix	0.86	0.81	8
	B	Brain	0.84	0.81	8	Prostate	0.22	0.15	10	Brain	0.85	0.84	8
9	M	Lymphoma	0.54	0.47	9	Lymphoma	0.73	0.63	9	Lymphoma	0.44	0.43	9
	F	Brain	0.80	0.76	9	Cervix	1.04	0.92	6	Brain	0.81	0.79	9
	B	Cervix	0.82	0.75	9	Leukemia	0.72	0.71	7	Cervix	0.75	0.72	9
10	M	Prostate	0.11	0.10	10	Brain	0.94	0.85	8	Prostate	0.07	0.06	10
	F	Leukemia	0.90	0.89	8	Brain	0.84	0.75	9	Leukemia	0.88	0.87	7
	B	Leukemia	0.95	0.91	7	Ovary	0.53	0.45	9	Leukemia	0.95	0.96	7

Notes: PR, PYLL Rate. SPR, SPYLL Rate. R, Ranks. Brain, Brain and nervous system. R^1^, ranks of high-risk cancers from Table 1 by region and gender. R^2^, ranks of high-risk cancer according to PYLL by region and gender.

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
