# Peer review of "Patterns of Life Lost to Cancers with High Risk of Death in China"

_ijerph, 2019, doi:10.3390/ijerph16122175_

Round 1
Reviewer 1 Report
Major comments:
In the "Materials and Methods" section, line 89, the authors write that "the survival age of E is 70 years old". Why was such an age limit accepted? In the article "High Cancer Burden in Elderly Chinese, 2005-2011" cited by the authors, different age limits are accepted. It will result in the inability to make comparisons.
In addition, in lines 91-92, the authors define N as "total population of the target population in 1-70 years old". Does it mean that deaths of children aged 0-1 years and people over 70 years do not increase the value of PYLL coefficients?
In lines 140-141, the authors write that the largest number of deaths due to cancers concerns the 70-79 age group. Therefore omitting this group appears to be a serious mistake affecting the reduction of the PYLL value.
In addition, if children 0-1 years old were not included in the study, why is the 0-4 age group given in the legend of charts 2 and 3?
Minor comments:
1. In line 96, the authors write that they chose to analyze the top 10 as the high-death-risk cancers in China. In the summary there are actually 10 cancer locations while in the Table 2 there are 13 locations. In the Results section, the authors refer to prostate cancers and cervical cancers also which take respectively the 11th and 13th place in the Table 2 list . Please harmonize the number of analyzed cancer locations.
2. In the Abstract section in lines 18-19 are listed the high-death-risk cancers in China. The third place is placed by gastric. In the Table 1 containing the ICD-10 codes stomach cancer is mentioned and the authors should use this name consistently throughout the all text.
3. In lines 220-221, the authors write "Moreover, the life lost due to lung, liver and breast cancers was 220 higher in urban areas; while that due to liver, lung, and gastric cancers was higher in rural areas. "This sentence is completely incomprehensible. Are lung and liver cancers higher in urban or in rural areas?
4. Charts 2 and 3 are illegible. The chart 1 is in color. Maybe charts 2 and 3 also would be colored?
5. In lines 209, 212 and 217, the authors refer to chart 4, which is not in the text of the manuscript. If I understand correctly it's about 3F, 3B and 3D charts?
6. In the Introduction section on line 51, the authors write about "unhealthy lifestyles in China". I think that this sentence should be clarified, including what lifestyle factors it concerns and what are the percentages of people affected by these risk factors.
Author Response
Response to Reviewer 1 Comments
Dear Reviewer:
Thank you for your careful review and kind comments.
We have already answered your comments (see below).
I hope you will take these responses into consideration and that they will answer your questions accurately.
Comments and Suggestions for Authors
Major comments:
Point 1: In the "Materials and Methods" section, line 89, the authors write that "the survival age of E is 70 years old". Why was such an age limit accepted? In the article "High Cancer Burden in Elderly Chinese, 2005-2011" cited by the authors, different age limits are accepted. It will result in the inability to make comparisons.
Re: Thank you very much for your careful review and questions. We calculated PYLL on the basis of references[1]. They limited the L of PYLL to 70 and only calculated the population aged 1-70. But after considering your doubts and the references you mentioned, we redefined L and recalculated PYLL based on life expectancy in China, the age ranged from 0 to life expectancy. Detailed calculation methods have been described in the paper, see the blue font in the method section. I hope this modification could meet your requirements.
[1] Romeder J M, Mcwhinnie J R. Potential Years of Life Lost Between Ages 1 and 70: An Indicator of Premature Mortality for Health Planning. International Journal of Epidemiology 1977, 6(2):143-151.
Point 2: In addition, in lines 91-92, the authors define N as "total population of the target population in 1-70 years old". Does it mean that deaths of children aged 0-1 years and people over 70 years do not increase the value of PYLL coefficients?
Re: It's a great honor to have your earnest check and reasonable query. According to your suggestion, we recalculated the PYLL. Detailed instructions were presented in the paper. Please review it again and hope it meets your standard.
Point 3: In lines 140-141, the authors write that the largest number of deaths due to cancers concerns the 70-79 age group. Therefore omitting this group appears to be a serious mistake affecting the reduction of the PYLL value.
Re: Thank you very much for your question. It's very scientific and reasonable. PYLL differed from mortality, according to its formula, it mainly reacted to the premature death (death earlier than life expectancy). Based on your suggestion, we recalculated the PYLL. Detailed instructions were presented in the paper. Please review it again and hope it meets your standard.
Point 4: In addition, if children 0-1 years old were not included in the study, why is the 0-4 age group given in the legend of charts 2 and 3?
Re: Thank you very much for the problems and mistakes you have found. We are very sorry and self-reproached for such mistakes. We have established a new calculation method for PYLL, including the 0-year-old group, to ensure the consistency of the whole article. I hope this will solve your problem.
Minor comments:
Point 1: In line 96, the authors write that they chose to analyze the top 10 as the high-death-risk cancers in China. In the summary there are actually 10 cancer locations while in the Table 2 there are 13 locations. In the Results section, the authors refer to prostate cancers and cervical cancers also which take respectively the 11th and 13th place in the Table 2 list . Please harmonize the number of analyzed cancer locations.
Re: Thank you very much for your careful observation. But let's make some explanations. Table 2 lists the top ten deaths of different regions and genders. However, due to geographical and gender differences, the same type of cancer ranks differently among different populations. For example, prostate cancer ranks 11 in the death order of all regions, but it ranks 8 in urban areas, so it appears in the results section. I'm sorry for the trouble caused by our table. I hope this explanation can answer your questions.
Point 2: In the Abstract section in lines 18-19 are listed the high-death-risk cancers in China. The third place is placed by gastric. In the Table 1 containing the ICD-10 codes stomach cancer is mentioned and the authors should use this name consistently throughout the all text.
Re: Thanks a lot for your meticulous examination. This is our mistake, and we have revised it to maintain the consistency of the full text.
Point 3: In lines 220-221, the authors write "Moreover, the life lost due to lung, liver and breast cancers was higher in urban areas; while that due to liver, lung, and gastric cancers was higher in rural areas. "This sentence is completely incomprehensible. Are lung and liver cancers higher in urban or in rural areas?
Re: We apologize for the unclear expression. We have revised and improved this part. Please review it again, hoping to meet your requirements.
Point 4: Charts 2 and 3 are illegible. The chart 1 is in color. Maybe charts 2 and 3 also would be colored?
Re: Thank you very much for your advice. We have colored all the charts and hope to meet your requirements.
Point 5: In lines 209, 212 and 217, the authors refer to chart 4, which is not in the text of the manuscript. If I understand correctly it's about 3F, 3B and 3D charts?
Re: Thanks a lot for pointing out the mistakes in this article. We are very sorry. We have revised it. Please review it again.
Point 6: In the Introduction section on line 51, the authors write about "unhealthy lifestyles in China". I think that this sentence should be clarified, including what lifestyle factors it concerns and what are the percentages of people affected by these risk factors.
Re: It's a great honor to receive your suggestion. We have made amendments in the paper and welcome to review it again.
Thank you very much!
If you have any questions, please don't hesitate to raise them. We will cooperate actively!
Thank you again!
Sincely yours,
Yi-zhong Yan

Reviewer 2 Report
Majors
1. The authors' statement in abstract: "The PYLL for lung, gastric, and esophageal cancer was 60–69 years" really puzzled me.
PYLL is defined as the difference between mean ages at death of the general population and those of lung cancer patients. If the average age at death is 75, how can PYLL be 60-69? I do not think the author calculate PYLL correctly.
2. Also in abstract: "The life lost of cancers had significant regional and gender differences".
In author’s PYLL formula, it seems that the overall life expectancy was used as the reference. However to compare different regionals and genders, the respective life expectancy should be used. The authors should specify these references specifically.
3. The authors did not explain their definitions of urban and rural. Is that by tier 1,2,3 or other standard? What’s the population odd of urban to rural according to the definition?
4. In table 2, the authors compared the SMR of China urban with that of world urban (as well as rural). What’s the reference of the world’s SMR in urban and rural? Are their definitions of urban and rural same as author’s?
5. According to previous studies [ref 8,9], “China has a greater cancer-related mortality as compared with developed 52 countries.” However in this study, the SMR of China and the world are roughly the same in both urban and rural with respect to each single cancer. What’s the implications of the inconsistency and the amazing coincidence?
6. It is difficult to catch the focus points among so many numbers in this paper. The authors should highlight the major new findings. The Conclusions section does not help.
Minors
1. Table 2, the SMR and CMR are "per 100,000 persons" should be stated.
2. "226,494,490 people (111595772 for urban, 114898718 for rural), accounting for 16.65% of the population."
What's the respective proportions of population for urban and rural?
Author Response
Response to Reviewer 2 Comments
Dear Reviewer:
Thank you for your careful review and kind comments.
We have already answered your comments (see below).
I hope you will take these responses into consideration and that they will answer your questions accurately.
Comments and Suggestions for Authors
Majors
Point 1: The authors' statement in abstract: "The PYLL for lung, gastric, and esophageal cancer was 60–69 years" really puzzled me.
PYLL is defined as the difference between mean ages at death of the general population and those of lung cancer patients. If the average age at death is 75, how can PYLL be 60-69? I do not think the author calculate PYLL correctly.
Also in abstract: "The life lost of cancers had significant regional and gender differences".
In author’s PYLL formula, it seems that the overall life expectancy was used as the reference. However to compare different regionals and genders, the respective life expectancy should be used. The authors should specify these references specifically.
Re: Thank you very much for your scientific questions and reasonable suggestions. We have revised our inaccurate expressions. After considering your doubts and suggestion, we redefined L and recalculated PYLL based on life expectancy in China, and there are regional and gender differences in life expectancy. Detailed calculation methods have been described in the paper, see the blue font in the method section. I hope this modification could meet your requirements.
Point 2: The authors did not explain their definitions of urban and rural. Is that by tier 1,2,3 or other standard? What’s the population odd of urban to rural according to the definition?
Re: Thank you very much for your professional questions. Our data comes from the annual report of cancer registration in China, and the division between urban and rural areas also comes from it. In the annual report, the classification of urban and rural areas is based on China's national standard GB2260-2009-"Code of Administrative Districts of the People's Republic of China". It classifies cities above prefecture-level into urban areas, counties and country-level cities into rural areas.
Point 3: In table 2, the authors compared the SMR of China urban with that of world urban (as well as rural). What’s the reference of the world’s SMR in urban and rural? Are their definitions of urban and rural same as author’s?
According to previous studies [ref 8,9], “China has a greater cancer-related mortality as compared with developed countries.” However in this study, the SMR of China and the world are roughly the same in both urban and rural with respect to each single cancer. What’s the implications of the inconsistency and the amazing coincidence?
Re: First of all, we apologize for the confusion caused by our incomplete expression. But allow us to make some explanations. The SMR of China and the world (SMRC and SMRW) in Table 1 refers to the standardized mortality rate based on the population structure of China and the world (we have explained it in the method section 2.4). It is calculated from the original crude mortality rate, not from other references. The purpose of the calculation is to facilitate comparisons with other studies at home and abroad, because mortality comparisons need to exclude the effects of different demographic structures in order to be comparable. In addition, for a disease, CMR, SMRC and SMRW do not differ much.
Point 4: It is difficult to catch the focus points among so many numbers in this paper. The authors should highlight the major new findings. The Conclusions section does not help.
Re: Thank you very much for your questions and suggestions. In order to provide a more comprehensive description, this study analyzed the life lost caused by cancers of different genders and ages in different regions in China, and including 10 types of cancers. So the results seem to be more complicated, but we have perfected them according to your suggestions, aim to make them more well-arranged and prominent. I hope this revision will meet your requirements.
Minors
Point 1: Table 2, the SMR and CMR are "per 100,000 persons" should be stated.
Re: Thank you very much for your advice. We have added the explanations in the notes of Table 1.
Point 2: "226,494,490 people (111595772 for urban, 114898718 for rural), accounting for 16.65% of the population." What's the respective proportions of population for urban and rural?
Re: Thank you very much for your question. The respective proportions of population for urban and rural were 16.77% and 17.04%. We have supplemented in the paper. Thank you.
Thank you very much!
If you have any questions, please don't hesitate to raise them. We will cooperate actively!
Thank you again!
Sincely yours,
Yi-zhong Yan

Round 2
Reviewer 2 Report
The authors had made substantial improvements to the article point by point.